# Extending the use of the World Health Organisations' water sanitation and hygiene assessment tool for surveys in hospitals – from WASH-FIT to WASH-FAST

Michuki Maina[1,2]*, Olga Tosas-Auguet[3], Jacob McKnight[3], Mathias Zosi[1], Grace Kimemia[1], Paul Mwaniki[1], Arabella Hayter[4], Margaret Montgomery[4], Constance Schultsz[2,5], Mike English[1,3]

**1** Health Services Research Group, KEMRI-Wellcome Trust Research Programme, Nairobi, Kenya, **2** Amsterdam University Medical Centres, University of Amsterdam, Amsterdam, The Netherlands, **3** Nuffield Department of Medicine, University of Oxford, Oxford, United Kingdom, **4** Water Sanitation and Hygiene Department, World Health Organization, Geneva, Switzerland, **5** Amsterdam Institute for Global Health and Development, Amsterdam, The Netherlands

* mmaina@kemri-wellcome.org

**Data Availability Statement:** All summary data underlying the findings are freely available in the

## Abstract

### Background

Poor water sanitation and hygiene (WASH) in health care facilities increases hospital-associated infections, and the resulting greater use of second-line antibiotics drives antimicrobial resistance. Recognising the existing gaps, the World Health Organisations' Water and Sanitation for Health Facility Improvement Tool (WASH-FIT) was designed for self-assessment. The tool was designed for small primary care facilities mainly providing outpatient and limited inpatient care and was not designed to compare hospital performance. Together with technical experts, we worked to adapt the tool for use in larger facilities with multiple inpatient units (wards), allowing for comparison between facilities and prompt action at different levels of the health system.

### Methods

We adapted the existing facility improvement tool (WASH-FIT) to create a simple numeric scoring approach. This is to illustrate the variation across hospitals and to facilitate monitoring of progress over time and to group indicators that can be used to identify this variation. Working with stakeholders, we identified those responsible for action to improve WASH at different levels of the health system and used piloting, analysis of interview data to establish the feasibility and potential value of the WASH Facility Survey Tool (WASH-FAST) to demonstrate such variability.

### Results

We present an aggregate percentage score based on 65 indicators at the facility level to summarise hospitals' overall WASH status and how this varies. Thirty-four of the 65

manuscript and supplemental files. The raw data used for this manuscript are hosted in a public repository Harvard Data verse. DOI Information: https://doi.org/10.7910/DVN/IJUWWR

**Funding:** MMa, GK, JM, M.Z and OT were supported by funds through a grant from the Economic and Social Research Council ESRCS # ES/P004938/1 awarded to ME. A Senior Research Fellowship awarded to ME by The Wellcome Trust (#207522) supported P.M. M Maina received additional support from a grant to the Initiative to Develop African Research Leaders (IDeAL) through the DELTAS Africa Initiative [DEL-15-003], an independent funding scheme of the African Academy of Sciences (AAS)'s Alliance for Accelerating Excellence in Science in Africa (AESA) and supported by the New Partnership for Africa's Development Planning and Coordinating Agency (NEPAD Agency) with funding from the Wellcome Trust [107769/Z/10/Z] and the UK government. The funders had no role in study design, data collection and analysis, decision to publish, or preparation of the manuscript.

**Competing interests:** The authors have declared that no competing interests exist.

indicators spanning four WASH domains can be assessed at ward level enabling within hospital variations to be highlighted. Three levels of responsibility for WASH service monitoring and improvement were identified with stakeholders: the county/regional level, senior hospital management and hospital infection prevention and control committees.

## Conclusion

We propose WASH-FAST can be used as a survey tool to assess, measure and monitor the progress of WASH in hospitals in resource-limited settings, providing useful data for decision making and tracking improvements over time.

## Introduction

Improving water supply, hygiene, sanitation and health care waste management (segregation, collection, disposal and treatment of health care waste) collectively abbreviated as WASH is a significant focus of the sustainable development goals and the global health agenda [1]. In health care facilities this improvement is linked to specific benefits. These include reductions in hospital-associated infections, antimicrobial resistance, better management and control of disease outbreaks, improved staff morale and an overall reduction in health care costs [2] [3]. The improvements in WASH also have a positive influence at the community level as health staff model proper hygiene practices even at the community level [4]—and may improve patients' trust in and experience of care and subsequently their satisfaction with and uptake of health services [5] [6].

Gains to improve WASH in health care facilities mainly in the low- and middle-income countries have been slow in the last decade. In 2015, the World Health Organization (WHO) and the United Nations Children's Fund (UNICEF) through the Joint Monitoring Programme for Water Supply, Sanitation and Hygiene highlighted some the current gaps with WASH. From this evaluation of about 60,00 health facilities, almost 40% of these health care facilities did not have access to an improved water source, about a third of them also lacked water and soap for handwashing and more than half lacked arrangements for safe disposal of health care waste [7]. A majority of these facilities assessed were in low- and middle-income countries.

In response to these challenges, the WHO/UNICEF developed the WASH in health care facilities global action plan to "*achieve universal access to WASH in all facilities in all settings by 2030*" [4]. As part of this initiative, core and extended indicators to track and improve WASH in health care facilities were developed, revised and tested across several African and Asian countries. The"Water and Sanitation for Health Facility Improvement Tool" (WASH-FIT) which contains these indicators was then developed [8]. This tool has been validated in several countries and was developed through consultation with experts and stakeholders before its eventual roll out[9]. It is mainly targeted at facilities in resource-limited settings. WASH-FIT covers four broad domains (Fig 1) and comprises 65 indicators and targets for achieving minimum standards for maintaining a safe and clean environment. These minimum standards are as set out in the WHO Essential environmental health standards in health care[10] and the WHO guidelines on core components of infection prevention and control programmes at the national and acute health care facility level [11].

WASH FIT was not designed for national or regional level situation analysis, monitoring or tracking of WASH in health care facilities. Instead, the tool guides health care facilities staff through a continuous cycle of assessing and prioritizing risks linked to poor WASH, defining

---

**Domains and related Sub-domains**

1. Water

2. Sanitation and healthcare waste

    o   Sanitation

    o   Healthcare Waste

3. Hand hygiene, environmental management, cleanliness and disinfection

    o   Hand Hygiene

    o   Environment, cleanliness and disinfection

4. Organisational management

---

**Fig 1. Domains assessed in WASH FIT[8].**

and implementing improvements and continually monitoring progress locally and autonomously. WASH FIT thus focuses on actions involving maintenance and repair as well as infrastructural and behavioural change, which are ideally integrated into broader quality improvement plans.

WASH FIT is meant to be adaptable to the local context but was initially developed for use in relatively small/less complex primary health care facilities providing outpatient services, family planning, antenatal care and maternal, newborn and child health services (including uncomplicated delivery; e.g. health centres, health posts and small district hospitals). Following inspection of the facility as a whole, WASH FIT involves scoring all 65 indicators using a three-level qualitative system (meets, partially meets, or does not meet the required standard), but it does not generate an overall hospital score nor can be used to generate a score for a particular service area or WASH domain.

Larger facilities (e.g. referral hospitals), however, raise specific issues. They deliver both inpatient and outpatient care spread across multiple wards, departments and service areas and they also have more complex management and leadership arrangements[8]. WASH FIT does not sufficiently consider the broader health system context and its potential for influencing local change.

In Kenya for example, in larger hospitals, the hospital health management team comprising the medical superintendent, health administrative officer, nursing officer in charge and the departmental heads are involved in the day to day running of the hospital [12]. These teams are assisted by different hospital committees constituted within the hospitals; these include infection prevention and control (IPC) committees. The hospital managers and committees prepare budgets and staffing needs, but the final budgetary and human resource allocation to these hospitals is the prerogative of regional/county government [12]. Majority of these larger hospitals in many low- and middle-income countries have similar organisational arrangements and some similar form of regional administration who have a role in decision making and resource allocation and need to be involved in the improvement of WASH.

Our report describes an adaptation of WASH-FIT to a Water Sanitation and Hygiene Facility Survey Tool (WASH FAST). This entails an extension of the tool to provide a comprehensive assessment of WASH services in hospitals providing both outpatient and inpatient care. It also provides a mechanism to meet both local, national and regional needs for tracking WASH

improvements. The adapted tool also considers the complex leadership and management arrangements. It proposes how responsibilities should be allocated across different levels of the health system to promote accountability and subsequent improvement.

## Methods

### Ethics statement

For this study, we sought and received informed consent in all cases where this was relevant. All information received was handled confidentially. All quotes from the study respondents were anonymised. This study received approval from the Oxford Tropical research ethics committee (OXTREC) from the University of Oxford (Ref: 525–17) and from the Kenyan Medical Research Institute (Ref: KEMRI/SERU/CGMR-C//086/3450).

### Adaptation of WASH-FIT into WASH-FAST

The adaptation of WASH-FIT into WASH-FAST entailed: (1) Creating an intuitive aggregation approach for the WASH indicators, to illustrate variation across health care facilities and facilitate tracking of WASH over time; (2) Extending assessment so that indicators are scored for each ward in addition to the facility as a whole—to highlight potential variation in WASH within a larger facility and; (3) identifying those responsible for action on WASH with relevant stakeholders. We illustrate the value of extending WASH-FIT to WASH-FAST by illustrating how data can be used for identifying challenges and highlighting variation.

**1. Aggregate scoring approach.** The WASH-FIT already presents a 'scoring' approach with one of three possible outcomes for each indicator, does not meet target, partially meets target and fully meets target. The first step involved moving from this qualitative scoring system to a simple quantitative scoring system that assigns a numeric score to each indicator based on assessment findings as follows: 0- does not meet the required standards (i.e. target), 1- partially meets target and 2- fully meets the target. This enabled us to create aggregate domain scores (based on the number of indicators within a domain) and aggregate facility scores (based on all 65 indicators) that can be used to show domain and facilities' performance. These aggregate scores can also be colour coded to produce an easy to interpret "traffic light" reporting approach.

**2. Identification of ward level indicators.** The second step involved identifying which of the existing and 65 WASH indicators can be assessed at the inpatient-ward level. To select indicators for assessment in every ward we employed an iterative process to review and discuss the 65 indicators involving the research team and a team of 19 health professionals comprising doctors, nurses, pharmacists and public health officers who had been recruited to pilot test and apply the WASH assessments in hospitals in Kenya. Using the same simple numeric scoring approach to the identified indicators as in step 1 above enables aggregate ward scores to be calculated to help identify variation between wards in the same hospital.

**3. Assigning responsibility for action.** The third part of the adaptation was to group indicators based on who should take responsibility for action to improve WASH–addressing the issue of accountability. For this process, a study team of 4 members familiar with the Kenyan health care system and its management examined all 65 indicators in a bid to understand how these indicators relate to one another and assign them to domains linked to the persons/offices who would be responsible for action to improve WASH. These levels of responsibility were confirmed through a series of interviews with health care workers and a subsequent large stakeholder workshop.

## Demonstrating potential and creating tools to help visualise performance and its variation

We proceeded to collect data using the WASH-FAST tool as part of a survey in 14 county hospitals varying in size and bed capacity across 11 counties during which key informant interviews were also conducted (see below). This survey is described in more detail in an accompanying paper [13]. In brief, the county hospitals included are in high and low malaria zones in Kenya (five and nine sites, respectively). The selection of these hospitals was purposeful and based on links developed from ongoing work to improve clinical information as part of a collaboration between the Kenya Medical Research Institute -Wellcome Trust Research Programme and the Ministry of Health [14]. The survey involved assembling a team of 7–8 people and conducting a facility assessment at each hospital. The study team included a leader, four surveyors employed for the study and 2–3 representatives selected based on their specific role as infection prevention and control coordinators or public health officers from the individual hospitals where the survey was being carried out. Data collection used the same methodology as WASH-FIT and involved direct observation and discussion with relevant hospital workers to provide clarification of the assessment where needed. Each indicator was assessed, and the score determined by team consensus as either not meeting target, partially or fully meeting the target. Data were collected for each inpatient ward (using 34 WASH indicators), then for indicators assessed at the whole facility level (65 WASH indicators). The 65 facility level indicators included an assessment of outpatient areas, common service areas (e.g. kitchen, laundry, laboratory, waste management facilities) and the outdoor environment, taking account of ward-specific scores where relevant, and represents an overall judgement of the survey team. The data collection tools and standard operating procedures used are provided in the supplementary information. (S1 File)

Aggregate scores were generated by summing individual indicator scores and dividing this total by a denominator that assumed a perfect score for each indicator. In this way, we then estimate percentage scores for the hospital, WASH domain and level of accountability using indicators linked to these grouping categories as appropriate. Summary ward-specific scores were based on individual indicator assessments made for each ward. The rationale for such sub-scores was to highlight variation and priority areas for improvement and who should take responsibility for improvement. To promote the rapid interpretation of scores, we generated 'traffic-light' colour maps presenting percentage scores using cut-offs of <40%, 40–60%, 60–80% and 80–100%. Data analysis for visualisation was done using R, an open-source statistical package [15].

## Use of qualitative data

Qualitative interviewing pursued two purposes; to understand IPC arrangements in Kenyan hospitals (with findings reported elsewhere) and to explore the feasibility and potential value of our proposed allocation of indicators for accountability.

The interviews were conducted with 17 hospital managers (e.g. medical directors, nursing and laboratory heads) and 14 frontline health workers (e.g. consultants, medical and nursing officers) during the survey visits, in seven of the 16 hospitals–sampled to ensure spread across different geographical locations represented by the study hospitals.

Interviews were conducted by the first, third, and fifth authors and took between 30 and 90mins. The first author led this section of work and followed a semi-structured interview approach. The interviews were generally guided by the 'long' or 'ethnographic' approach [16], but there was a particular focus on responsibility for different areas of IPC and WASH, which provided more structure to the inquiries in this area. All authors have experience with medical

research in Kenya, but the first author is a well-experienced doctor with experience working in different county structures, and he guided the other interviewers over the course of the interviewing.

The interviews were prefaced by an explanation from a senior member of staff who had given permission for the research to take place on-site. All interviewees were then given time to read a background information sheet concerning the project, and each signed a full written consent. An opportunity was offered to critique or refuse the interview, or to withdraw permission, but no respondent chose to do this.

We used both purposive and snowball sampling in order to identify respondents, and as much as was practically possible, we were mindful of the mix of gender, age and experience and aimed to reflect this diversity in our interviewee sampling strategy. Each interviewee was introduced to the researchers by a senior member of staff familiar with the research and the interview took place in or near their place of work. The interviews were conducted with one or two researchers away from patients and staff. We ensured in each case that we did not take the respondent away from core tasks or risk harm to their patients.

No repeat interviews were felt to be necessary, but the interview instrument was honed to focus on areas of interest over time, allowing us to move beyond areas where we had reached saturation and onto other new areas. The audio files were transcribed and uploaded into NVivo 12 and the audio files were kept on an encrypted laptop. It was relatively trivial to complete our primary goal of identifying the formal, de jure responsibilities for each level, but it was also important to code and describe the nuances of the de facto practices that prevail in the studied sites (as described below).

We did not return transcripts to the respondents, but we have sought to share general findings with hospital management and through the ministry and county stakeholders with whom we are connected. Most importantly, we used a stakeholder consultative workshop to confirm or revise the levels of accountability and related indicator sets. This workshop included approximately 120 technical experts and key stakeholders in WASH comprising; Ministry of Health officials, Hospital WASH leaders, county health department leaders, and doctors and nurses in Kenya with interest in IPC. This cross-checking of indicator allocation to different levels of responsibility was completed before creating scores for these domains. We also used the stakeholder workshop to get feedback on the use of aggregate scores and data visualisation approaches, confirming that the proposed reporting methods would be of value to potential end-users.

The interview guides used for the study are available as a supplement. (S2 File) A (Consolidated criteria for Reporting Qualitative research) COREQ checklist was successfully completed and is included as an appendix. (S3 File)

## Results

From the tool redesign to collect data at ward level, we established that 34 of the 65 indicators could also be assessed at the ward level. A description of all 65 indicators is provided as a supplement. Table 1 below provides a summary of the number of indicators that were to be assessed at the ward and facility level by the original WASH-FIT domains and by the proposed levels of responsibility.

### Responsibility for action

We developed a re-organisation of the existing WASH indicators based on their logical relationship and who would be responsible for action resulting in a classification with three levels of responsibility. These are, first, the county government which should be concerned with

**Table 1. Summary indicators at ward and facility level by WASH domains and WASH-FAST.**

| WASH-FIT | | WASH-FAST | | | | | | |
|---|---|---|---|---|---|---|---|---|
| WASH DOMAINS | | WASH DOMAINS | | | ACCOUNTABILITY DOMAINS | | | |
| | Facility | | Ward | Facility | | Ward | Facility | |
| Water | 14 | Water | 6 | 14 | County Government | 0 | 9 | |
| Sanitation & Health care Waste | 22 | Sanitation & Health care Waste | 11 | 22 | Hospital Management | 16 | 31 | |
| Hand hygiene, Environmental Management, Cleanliness and Disinfection | 18 | Hand hygiene, Environmental Management, Cleanliness and Disinfection | 12 | 18 | Infection prevention & control committee | 16 | 25 | |
| Organisational Management | 11 | Organisational Management | 5 | 11 | | | | |
| **Total** | **65** | **Total** | **34** | **65** | **Total** | **32** | **65** | |

indicators that are beyond the control of hospital leadership (this level might be a national government where resources are not fully devolved). The second level is the hospital health management team (the medical superintendent, health administrative officer, the nursing officer in charge and the departmental heads) and the last level is the hospital infection prevention and control committee (Table 1).

On the proposed levels of responsibility in the WASH-FAST, although 2 of the 9 indicators under the responsibility of the county government could also be assessed at ward level, these are (i) water services available in sufficient amounts and (ii) having rewards for high performing staff, these only need to be assessed at the facility level for tracking progress in follow up assessments. Therefore, when grouped by the responsibility, we suggest only 32 of the original 34 indicators are assessed at ward level (Table 1).

The in-depth interviews allowed us to explore the relationships between the WASH criteria and to establish where responsibilities lay for each. This contributed to the emerging model of the layers of WASH management and informed our understanding of the causalities and contingencies in this area.

County Level–The County is responsible for setting the budget for each hospital, and importantly, sets the overall budget for health spending. This impacts on general, but hugely important, WASH-related criteria such as staffing levels and material upkeep of hospitals. Additionally, while each department in each hospital is asked to project their needs for the next year as part of hospital budgeting processes, the requested amounts may be ignored by counties. Hospitals thus needed to work within the limitations of the budget and staffing allowed them.

*"You know normally we are told to itemise whatever we require in the departments that we are working in . . .yes, by different departments, come up with their budget proposal. The administrator compiles the budget for the whole hospital and then give it. . .we don't control funds in the institution. Every finance that is channelled to the hospital is controlled by the chief officer in the county. So, we send the budget to the county"* <u>Hospital Manager</u>

Whereas the day to day running of the hospital is done by the hospital management, some of the activities are delegated to committees within the hospital.

Hospital Level–Key areas of hospital management were in part delegated to committees that held responsibilities for activities and addressing needs. The effectiveness of IPC committees in different hospitals appeared to be variable, but where they were operational, they had an essential influence on resource allocation and monitoring of WASH.

*". . .when the committee, the IPC committee, meets they raise their needs as per various departments, and then the hospital now addresses that. Like if you want to purchase, for*

*instance, you want bins, litter bins, disposal bags, waste disposal bags. So, you raise your needs as per your department because you know different departments have got their different needs"* <u>Hospital Manager</u>

However, despite their importance, these IPC committees in some facilities struggled to gain respect relative to other more prestigious committees and were regarded to be of low status.

*There are some committees which are found to be more, which are more do I say prestigious? They look better. So, if I am in IPC, people will be thinking okay. . . so IPC will have no one. I mean, what is the benefit of being in IPC, what is there, how am I gaining being in IPC?* <u>Consultant</u>

Ward Level–Interest and capability at the ward level is essential to effective WASH. The individuals responsible for WASH at this level are not likely to have the ability to affect budgets and resource allocation, but they are essential in both maintaining supplies and overseeing important areas such as hand hygiene. Variability in performance at ward level may be linked to the presence of an individual in the ward who has interest and passion for IPC related activities.

*"And we also have someone, he's also a team leader in the infection control and making sure we have . . . whenever he's available we have our sanitizers, make sure we have soap, make sure we have gloves"* <u>Frontline Health Worker</u>

The relative importance of IPC varied ward to ward; however, with the newborn units (NBU) often used as an example to contrast high versus low performance:

*"Across the hospital, in NBU is where I know there is strict infection prevention because once you are getting into NBU, you remove your lab coat, you wash your hands and then you get into the unit where you fold whatever you are wearing, a long-sleeved anything you fold it, and then you get in the unit. . . Now in other wards, we don't have such strict infection prevention, you get in, and you start . . ."* <u>Frontline Health Worker</u>

### Consultative workshop

The consultative workshop was held in November 2018 during the annual national IPC symposium. There were 120 people in attendance. These included Ministry of Health officials, managers from the hospitals and county government, development partners and training institutions who are familiar with infection prevention and control and WASH and frontline health workers. The workshop attendees discussed, amended and approved the proposed levels of accountability (Table 1) and made specific recommendations that hospitals identify a champion to lead the IPC committees and to identify ways of boosting morale for IPC related issues among health workers across these hospitals.

Based on all of this work, a final indicator framework was developed (Fig 2) that shows the relationship between the indicators, their original WASH-FIT domains, and how they are allocated to different levels of responsibility. We also use Fig 2 to highlight which of the individual indicators can also be assessed at ward level.

### Visualisation approaches to support monitoring

Using an example of data collected from four of the 14 hospitals, two large (H2, H9) and two small (H1, H7) hospitals, we present an illustration (Fig 3) of how performance of two

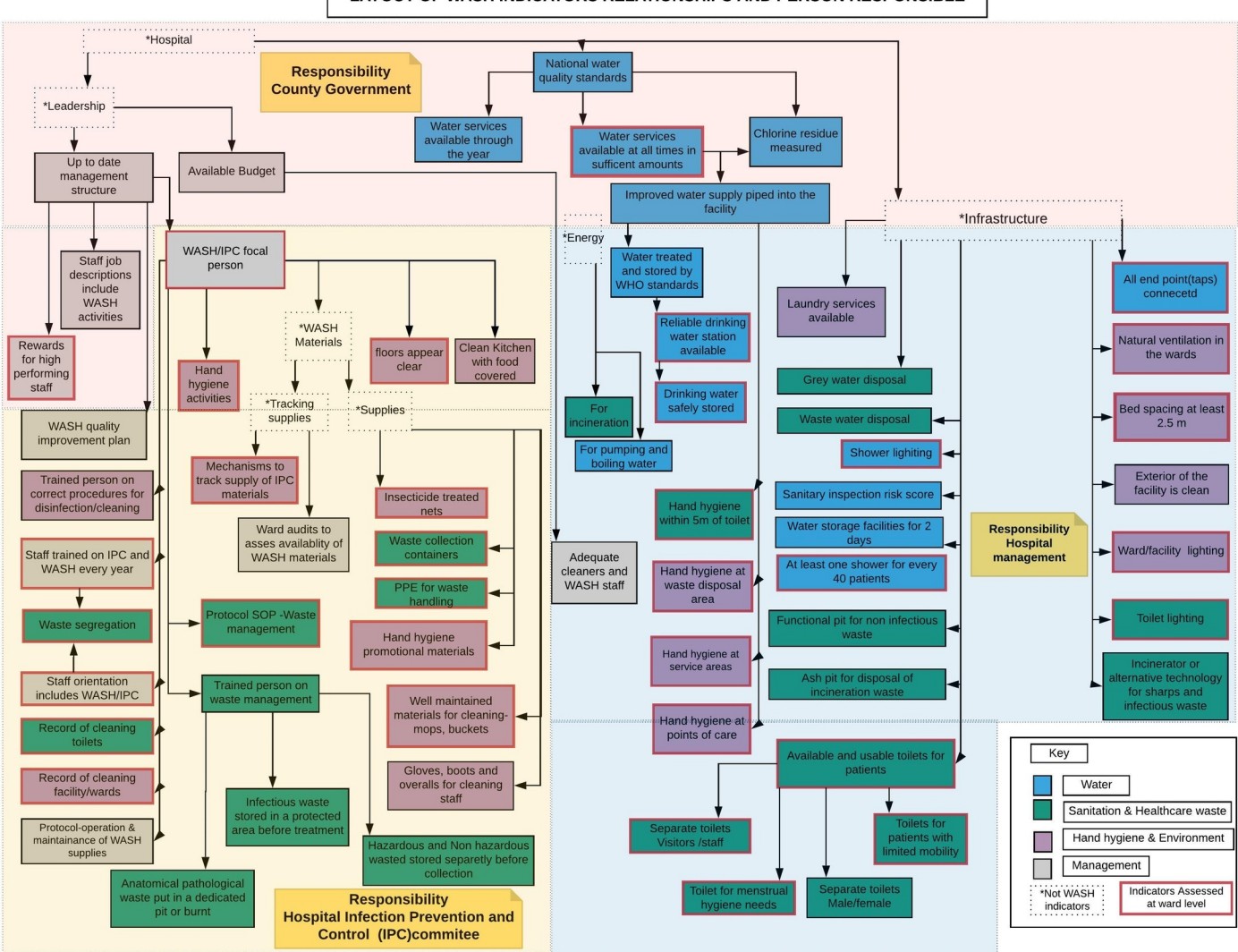

**Fig 2. Schematic layout of WASH FIT indicators.** Illustrates how indicators assessed at ward and facility level are logically related. These are grouped by the original 4 domains and by levels of responsibility. The indicators with a red bold outline were also assessed at ward level. The dotted boxes are used to describe categories and are not part of the indicators.

domains (water and sanitation) vary between hospitals (Panel A) and how the individual wards within these facilities performed (Panel B). We note differences between domains and differences between hospitals, with some facilities having scores of <50%. We also note variability across wards in these hospitals. From this example (Panel B) for the water domain ward scores in hospital H1 show, minimal variability compared to those of hospital H9. We contrast our visualisations with data presented using the original WASH-FIT template at the facility level for the four hospitals in Fig 3, Panel C.

To further illustrate how WASH-FAST can provide detailed information for use at national and regional levels on hospital performance and where responsibility for action lies, we present an example of all the 16 indicators [spanning all the WASH domains] under the IPC committee at ward level. Here we generate the summary ward scores for each of the four hospitals (H1, H2, H7, H9) coded using a traffic light colour system with red being a score of <40% and

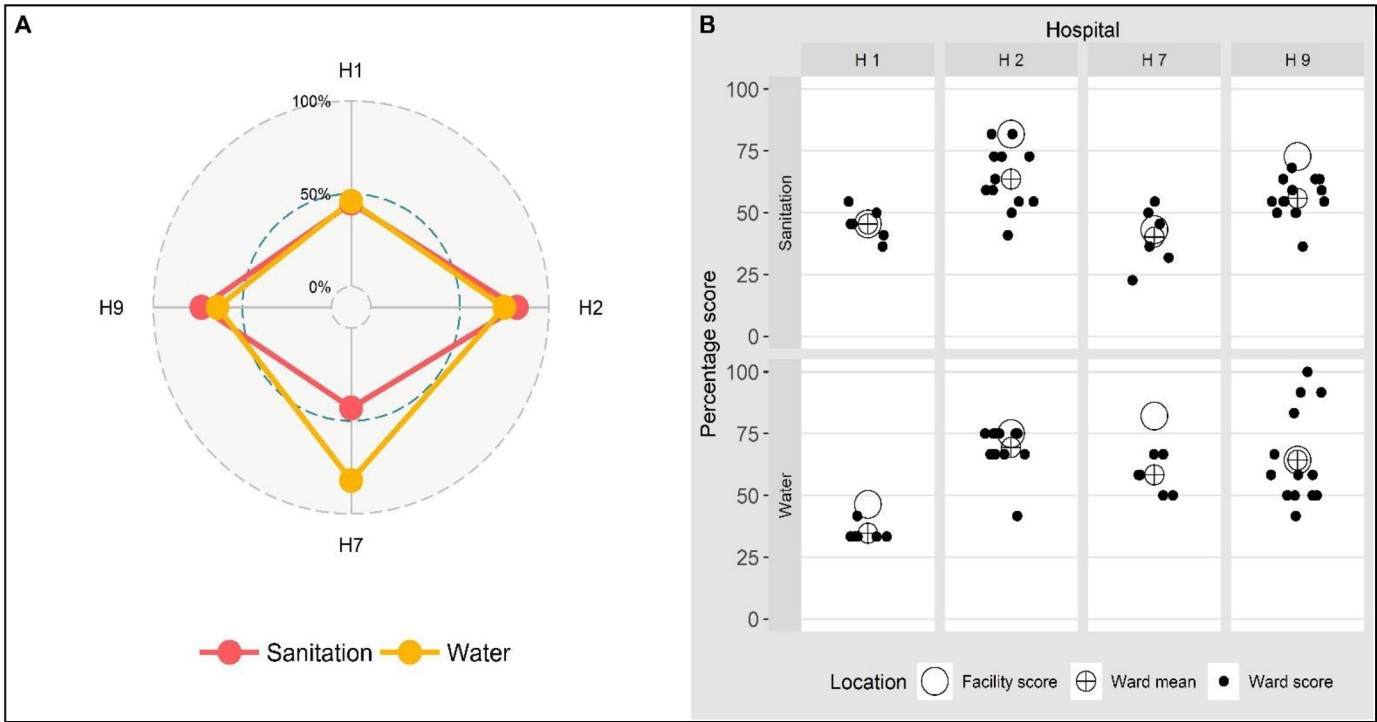

**Fig 3. Service performance variation by ward and hospital and the original WASH FIT scores.** Panel A: Radar Plot of facility-level scores from four hospitals for two domains (Water, Sanitation) showing similar performance for sanitation overall but more marked variation for water varying hospital performance. Panel B: Shows ward domain scores from multiple wards (dots) for two domains (water, sanitation) illustrating their variation, the mean of these ward-specific scores (circled cross) and the overall facility aggregate score(plain circle). The overall facility score includes assessment of inpatient wards and other service areas (kitchen, outpatient, outdoor environment) across the hospital. Panel C shows WASH-FIT facility-level scores of four hospitals for the two domains.

| Domain (Number of indicators) | Hospital | Indicator scores | | |
|---|---|---|---|---|
| | | **Indicators meeting target** | **Indicators partially meeting target** | **Indicators not meeting target** |
| **Water** (14) | **H1** | 4 | 5 | 5 |
| | **H2** | 8 | 5 | 1 |
| | **H7** | 10 | 3 | 1 |
| | **H9** | 7 | 4 | 3 |
| **Sanitation** (22) | **H1** | 8 | 4 | 10 |
| | **H2** | 15 | 6 | 1 |
| | **H7** | 4 | 11 | 7 |
| | **H9** | 13 | 6 | 3 |

green indicating a score of >80%. Fig 4 illustrates a 'dashboard' approach that shows performance across the hospitals for the individual IPC-committee related indicators assessed at ward level (horizontal bar chart, for example highlighting a need for the IPC committees to

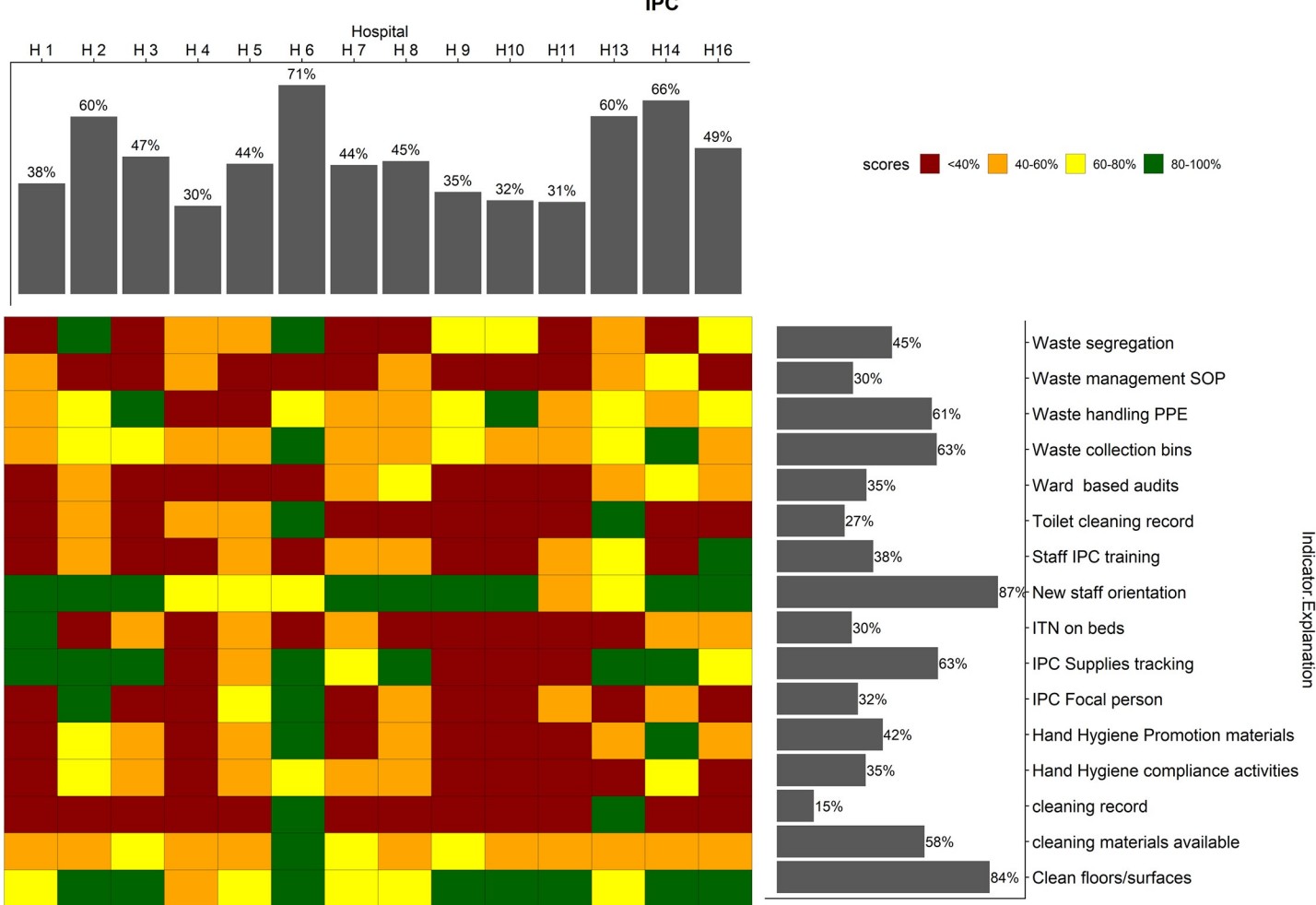

**Fig 4. Performance of Infection prevention and control domain indicators.** The mean service performance at ward level for the indicators under the IPC committee is shown by the upper bars. The right bars summarize the performance of each indicator across the 4 hospitals. The squares in the central grid are coloured according to the performance classification of each indicator in each hospital, by the colour categories. SOP: standard operating procedures, PPE: personal protective equipment; IPC: infection prevention and control; ITN: insecticide-treated nets.

work on availing cleaning records in the wards in these four hospitals). It also helps visualise the overall mean ward scores for each hospital for all 16 indicators (the top panel vertical bar chart). The central traffic light coding presents a summary of how the individual indicators performed in each specific hospital.

## Discussion

We have presented an adaptation of the Water and Sanitation for Health Facility Improvement Tool (WASH-FIT) into WASH-FAST (Facility Survey Tool). The adaptation entailed an extension of the tool to meet primarily national (i.e. situation analysis, monitoring and tracking) needs, and to facilitate comprehensive assessment of WASH services in larger–more complex–secondary and tertiary health care facilities s encompassing both outpatient and inpatient care and multiple medical specialties. The adapted tool scores indicators at various levels of the facility (including by ward and by medical specialty) and assigns levels of accountability for each indicator, to identify what services can be addressed by whom locally or at higher levels

of the health system. An aggregated numeric scoring system, consisting of a percentage score out of the total that would be obtained if all indicators met the expected target, can be used to identify service areas requiring priority action within a facility or to identify facilities or specialties requiring priority action nationally or sub-nationally.

Adequate governance and leadership are one of the foundations for the provision of quality care. Governance for quality includes improving accountability and identifying the roles and responsibilities at all levels of health systems and using data to make decisions [17]. Thus WASH-FAST may also be used to identify responsible actors limiting or effecting positive change within a facility and beyond, and to potentially reward excellent performance in a bid to improve staff morale concerning IPC/WASH. WASH-FAST assumes that performance and quality indicators for WASH are the responsibility of three possible actors. These are, administrative division officers or governments responsible for budgetary and human resource allocation to hospitals, senior hospital management teams and relevant facility-based specialised committees or groups of persons who are essential in decision making for IPC related activities, such as the infection prevention and control committee. Although the WASH-FAST was developed within a Kenyan context, we expect these broad accountability domains (endorsed by government representatives, public health officers, IPC experts and health care professionals through interviews and a consultative workshop), to be generally applicable to most low- and middle-income countries, with minor context-appropriate considerations. This would allow more comprehensive use of WASH-FAST and could support within or between-country comparisons where relevant.

We anticipate that aggregated scores derived from the application of WASH-FAST can be used more broadly to inform health system leaders on whether and what facilities and specialties require action at either local, sub-national or national level to improve WASH services. The simple scores allow the comparison of WASH services within and between facilities and or medical specialties either cross-sectionally or over time (i.e. to identify changes in quality and performance and trends), through repeated surveys. The extended tool is hence a broadly applicable facility improvement tool–potentially encompassing training, team-building and risk assessment steps as per the WASH-FIT process—that also appertains to WASH performance monitoring sub-nationally and nationally. Training of health care facilities staff to partake in surveys and facility improvement plans, in turn, empowers and encourages staff to take interest and ownership on WASH and IPC, contributes to up-skilling in these areas and improves short and long term sustainability of interventions and developments where applicable [18]. WASH-FAST may also be applied to help remedy the paucity of data on the status of WASH services in low- and middle-income countries, help bridge evidence-based gaps and provide a platform to monitor interventions aimed at improving AMR and patient safety. This is while continuing to serve the original purpose of continuously informing a local improvement plan in small primary health care facilities as well as more extensive facilities comprising multiple wards and medical departments.

A limitation of both WASH-FAST (and WASH-FIT), is that the score assigned to selected individual indicators may be subjective, where it relies on observations that could vary from person to person. To mitigate this, we developed standard operating procedures before data collection, conducted training for the data collection teams and used consensus among surveyors to assign scores during the assessments. WASH-FAST also rests on the premise that the hospitals have well-structured leadership, including a functioning infection prevention and control committee or relevant expert group. The indicators are also not weighted in accordance to the health risk they pose, implying that identical aggregate scores may have very different decision-making implications depending on the composite of indicators considered in the score (e.g. availability of water vs cleaning protocols). The same limitation applies to repeated measurement, where a facility may get the same score over two consecutive surveys,

perhaps reflecting improvements in some areas but worsening in others. To mitigate these limitations, aggregated (summary) scores should be interpreted in the context of individual indicator scorings presented through heatmaps or other visualisation tools.

We appreciate that although we adapted an already validated tool (WASH-FIT), there is need to extend the use of the WASH-FAST to other populations and settings to apply the tool under operational conditions, not by researchers.

Although it was not our aim to validate the WASH-FAST, we explored key elements of face validity as described by Nevo [19]. We involved stakeholders, experts and health workers in the consultative workshop, to check if the items in the WASH-FAST were appropriate (rater involvement). The research team and the stakeholders also assessed if the content in the tool, including the levels of responsibility, was valid (hypothesised validity)[19]. The other major part of face validity involves establishing if the method of measurement is appropriate, and for this, we established through the consensus in the workshop that the WASH-FAST was indeed suitable for assessing WASH practices. However, looking at the nature of the tool and similar tools, where there are no gold standard measures, validation may be challenging. It may need to be sufficient that there is a consensus on the value of addressing the issues identified by indicators. This may be thought of more as similar to an appraisal than a true measurement of WASH performance on a linear scale.

To improve hospitals as platforms that provide high-quality care and prevent the emergence of AMR, proper WASH and IPC structures are core [17]. We suggest that using WASH-FAST to monitor and improve the capacity for WASH and IPC would enhance governance for quality and limit the emergence of AMR by promoting accountability and identifying the roles and responsibilities at all levels of the health systems [17]. In the process of accelerating universal health coverage in many counties, hospital accreditation has become a key component as it provides for insurers and governments a criterion for which hospitals to include in their funding mechanisms. WASH-FAST can thus be used as part of the tools for hospital accreditation to ensure they focus adequately on IPC structures as part of preventing patient harm and AMR [20] [21].

## Conclusion

We propose the WASH-FAST (Survey) tool as a modification/extension of the original tool. Compared to the WASH-FIT, WASH-FAST provides for additionl assessment of WASH within the hospital and assigns responsibility for action. Its use is most relevant in larger hospitals that have multiple inpatient units(wards) as it allows for assessment at the ward level in addition to the overall facility assessment. Due to its ability to provide aggregate scores, its can be used to monitor and track the progress of WASH at hospital, regional or national levels providing crucial data for governments and international development agencies who provide support for WASH. In addition due to the ability to assign responisibilty for action, WASH-FAST allows for persons/teams to take responsibility in improving the state of WASH at hospital and regional level. Where the primary aim is to support local improvement in smaller facilities, WASH-FIT remains the tool of choice.

## Supporting information

**S1 File. WASH Data collection and Standard Operating procedures tool.**
(DOCX)

**S2 File. Qualitative interview guide.**
(PDF)

**S3 File. Consolidated criteria for reporting qualitative studies (COREQ): 32-item checklist.**
(DOCX)

## Acknowledgments

We thank the Ministry of Health and the council of governors who permitted this work to be carried out. We also thank the hospital management and clinical teams who supported the work in the survey hospitals. This work is published with the permission of the Director of KEMRI.

## Author Contributions

**Conceptualization:** Michuki Maina, Olga Tosas-Auguet, Jacob McKnight, Constance Schultsz, Mike English.

**Data curation:** Michuki Maina, Mathias Zosi.

**Formal analysis:** Michuki Maina, Olga Tosas-Auguet, Jacob McKnight, Grace Kimemia, Paul Mwaniki.

**Funding acquisition:** Olga Tosas-Auguet, Mike English.

**Investigation:** Michuki Maina, Olga Tosas-Auguet, Jacob McKnight, Mathias Zosi, Grace Kimemia.

**Methodology:** Michuki Maina, Olga Tosas-Auguet, Jacob McKnight, Grace Kimemia, Paul Mwaniki, Arabella Hayter, Margaret Montgomery, Constance Schultsz, Mike English.

**Project administration:** Mathias Zosi, Mike English.

**Resources:** Arabella Hayter, Margaret Montgomery.

**Software:** Grace Kimemia, Paul Mwaniki.

**Supervision:** Olga Tosas-Auguet, Jacob McKnight, Constance Schultsz, Mike English.

**Validation:** Olga Tosas-Auguet, Jacob McKnight, Arabella Hayter, Margaret Montgomery, Mike English.

**Visualization:** Michuki Maina, Olga Tosas-Auguet, Paul Mwaniki.

**Writing – original draft:** Michuki Maina, Olga Tosas-Auguet, Jacob McKnight, Mike English.

**Writing – review & editing:** Michuki Maina, Olga Tosas-Auguet, Jacob McKnight, Mathias Zosi, Grace Kimemia, Paul Mwaniki, Arabella Hayter, Margaret Montgomery, Constance Schultsz, Mike English.

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
