## [Decision Letter · Decision Letter 0]

3 Sep 2019

PONE-D-19-19422

EXTENDING THE USE OF THE WORLD HEALTH ORGANISATIONS’ WATER SANITATION AND HYGIENE ASSESSMENT TOOL FOR SURVEYS IN HOSPITALS – FROM WASH-FIT TO WASH-FAST

PLOS ONE

Dear Dr Maina,

Thank you for submitting your manuscript to PLOS ONE. After careful consideration, we feel that it has merit but does not fully meet PLOS ONE’s publication criteria as it currently stands. Therefore, we invite you to submit a revised version of the manuscript that addresses the points raised during the review process.

We would appreciate receiving your revised manuscript by Oct 18 2019 11:59PM. To enhance the reproducibility of your results, we recommend that if applicable you deposit your laboratory protocols in protocols.io, where a protocol can be assigned its own identifier (DOI) such that it can be cited independently in the future. For instructions see: http://journals.plos.org/plosone/s/submission-guidelines#loc-laboratory-protocols

We look forward to receiving your revised manuscript.

Kind regards,

Lars-Peter Kamolz, M.D., Ph.D., M.Sc.

Academic Editor

PLOS ONE

Journal Requirements:

1. Please include copies of the interview guide(s) used in the study, in both the original language and English, as Supporting Information, or include a citation if they have been published previously.

3. Thank you for including the following funding information within your acknowledgements section; "This work was supported by funds from the economic and social research council ESRCS # ES/P004938/1, and a Senior Research Fellowship awarded to ME by The Wellcome Trust (#097170). MM is supported by a grant from by the Initiative to Develop African Research Leaders (IDeAL) through the DELTAS Africa Initiative [DEL-15-003], an independent funding scheme of the African Academy of Sciences (AAS)'s Alliance for Accelerating Excellence in Science in Africa (AESA) and supported by the New Partnership for Africa's Development Planning and Coordinating Agency (NEPAD Agency) with funding from the Wellcome Trust [107769/Z/10/Z] and the UK 445 government. The funders had no role in drafting nor the decision for submitting this manuscript."

Reviewers' comments:

Reviewer's Responses to Questions

**Comments to the Author**

1. Is the manuscript technically sound, and do the data support the conclusions?

Reviewer #1: Partly

Reviewer #2: Yes

2. Has the statistical analysis been performed appropriately and rigorously? 

Reviewer #1: N/A

Reviewer #2: Yes

3. Have the authors made all data underlying the findings in their manuscript fully available?

Reviewer #1: Yes

Reviewer #2: Yes

4. Is the manuscript presented in an intelligible fashion and written in standard English?

Reviewer #1: No

Reviewer #2: Yes

5. Review Comments to the Author

Reviewer #1: Dear authors,

Thank you for the opportunity to review the manuscript, “EXTENDING THE USE OF THE WORLD HEALTH ORGANISATIONS’ WATER SANITATION AND HYGIENE ASSESSMENT TOOL FOR SURVEYS IN HOSPITALS– FROM WASH-FIT TO WASH-FAST.”

For the limited national context, possibly an interesting study that possibly is suitable for the national Health Department. Lacking scientific underpinning and low applicability for an international context. No considerable new insights.

Reviewer #2: Dear authors,

Thank you for the opportunity to review your manuscript "Purposeful design of a survey tool to evaluate the adequacy of hospitals'

water and sanitation and hygiene and allocate responsibility for action - From WASH FIT to

WASH FAST" , in which you present important limitations of WASH FIT and Adaption do WASH FAST.

Firstly, I would like to congratulate you for having drafted that very relevant and interesting research. However, prior a possible publication, I would like to share my thoughts on the manuscript:

1.) Within the Abstract the limitations of WASH-FIT should be highlighted shortly, to comprehend the need for an adaption.

2.) Please try and design the background part in a more legible way. There are too many abbreviations in quick succsession which makes it hard to understand.

Thank you

6. PLOS authors have the option to publish the peer review history of their article (what does this mean?). If published, this will include your full peer review and any attached files.

Reviewer #1: No

Reviewer #2: No

---

## [Author Response · Author response to Decision Letter 0]

1 Oct 2019

Thank you for the review and comments provided for this manuscript. We have addressed the one query raised as shown below. 

Reviewers Comments 

Reviewer #1: Dear authors,

Thank you for the opportunity to review the manuscript, “Extending the use of the World Health Organisations’ water sanitation and hygiene assessment tool for surveys in hospitals– from WASH-FIT to WASH FAST.”

For the limited national context, possibly an interesting study that possibly is suitable for the national Health Department. Lacking scientific underpinning and low applicability for an international context. No considerable new insights.

Response 

We thank the reviewers for taking the time to review this manuscript. In response to the above comment.

Water Sanitation and Hygiene (WASH) has been highlighted as a crucial but under-resourced aspect of the fight against hospital-acquired infections and antimicrobial resistance (AMR) and policymakers’ efforts to intervene in these areas are fundamentally limited. This has been in part due to the lack of understanding of the status of the hospitals under their charge. The WASH-FIT tool is in use around the world and the World Health Organization (WHO) has promoted its use as a tool in improving water and sanitation and thus reducing infection and opportunities for AMR. As such, our extension of this important tool is highly relevant to Low- and middle-income countries considering interventions in this important area and could increase the impact of the tool wherever it is used. This work is relevant to teams conducting research or developing interventions to improve WASH as it forms a good base to measure performance on WASH. For governments, international organizations and funders who are interested in investing in WASH activities, our work also provides a possible way to measure and monitor WASH performance. 

The original WASH FIT from which we propose the extension underwent a rigorous validation process and was piloted in several countries before its eventual rollout ( Page 4/5). The 65 WASH indicators used for assessment in the tool are based on existing scientific evidence and are derived from global environmental and infection prevention and control standards (Page 5 Line 76-80). We are therefore of the view that this work indeed has major scientific underpinning. We have also highlighted this in the discussion section of the manuscript. For the reviewer’s reference, we are actively working with the WHO to ensure our findings do have a global impact. 

Reviewers Comments 

Reviewer #2: Dear authors,

Thank you for the opportunity to review your manuscript "Purposeful design of a survey tool to evaluate the adequacy of hospitals' water and sanitation and hygiene and allocate responsibility for action - From WASH FIT to WASH FAST", in which you present important limitations of WASH FIT and Adaption do WASH FAST. Firstly, I would like to congratulate you for having drafted that very relevant and interesting research. However, prior to a possible publication, I would like to share my thoughts on the manuscript:

1.) Within the Abstract the limitations of WASH-FIT should be highlighted shortly, to comprehend the need for an adaption.

2.) Please try and design the background part in a more legible way. There are too many abbreviations in quick succession which makes it hard to understand.

Response 

Thank you for your review and feedback.

1. We have put in the abstract a sentence on the main limitations of the WASH FIT. Which are that it was mainly designed for smaller hospitals with mainly outpatient and limited inpatient care. It was also not designed as a tool to monitor or compare performance across hospitals (Page 2 Line 25-26). We have also highlighted these are other limitations more clearly in the introduction section of the manuscript. (Page 5 Line 80-96)

2. Our apologies for the several abbreviations, we have attempted to put most of them in the manuscript as full text instead. The introduction section of the manuscript has been modified to be clearer and more legible. 

The first section of the introduction highlights what Water Sanitation and Hygiene(WASH) is and its importance in health (Page 4 Line 51-59).

In the second section, we describe what is known on the subject and the existing gaps (Page 4 Line 61-68).

We then proceed to highlight the introduction of the tool developed by the World Health Organization to monitor and improve WASH (Page 4 Line 69-80).

The next paragraph goes on to highlight what are the main shortcomings of the WASH FIT tool as it stands (Page 5 Line 82-98).

Using Kenya as an example we present the context in which this tool is meant to work highlighting that the health systems are more complex than envisaged in the WASH tool. (Page 6 Line 99-107).

The final paragraph highlights what our work is about by introducing a modification of the tool to meet some of the shortcomings described earlier (Page 6 Line 108-114).

---

## [Decision Letter · Decision Letter 1]

25 Oct 2019

PONE-D-19-19422R1

Extending the use of the World Health Organisations’  water sanitation and hygiene assessment tool for surveys in hospitals – from WASH-FIT to WASH-FAST

PLOS ONE

Dear Dr Maina,

Thank you for submitting your manuscript to PLOS ONE. After careful consideration, we feel that it has merit but does not fully meet PLOS ONE’s publication criteria as it currently stands. Therefore, we invite you to submit a revised version of the manuscript that addresses the points raised during the review process.

We would appreciate receiving your revised manuscript by Dec 09 2019 11:59PM. To enhance the reproducibility of your results, we recommend that if applicable you deposit your laboratory protocols in protocols.io, where a protocol can be assigned its own identifier (DOI) such that it can be cited independently in the future. For instructions see: http://journals.plos.org/plosone/s/submission-guidelines#loc-laboratory-protocols

We look forward to receiving your revised manuscript.

Kind regards,

Lars-Peter Kamolz, M.D., Ph.D., M.Sc.

Academic Editor

PLOS ONE

Reviewers' comments:

Reviewer's Responses to Questions

**Comments to the Author**

1. If the authors have adequately addressed your comments raised in a previous round of review and you feel that this manuscript is now acceptable for publication, you may indicate that here to bypass the “Comments to the Author” section, enter your conflict of interest statement in the “Confidential to Editor” section, and submit your "Accept" recommendation.

Reviewer #1: (No Response)

Reviewer #2: All comments have been addressed

2. Is the manuscript technically sound, and do the data support the conclusions?

Reviewer #1: Partly

Reviewer #2: Yes

3. Has the statistical analysis been performed appropriately and rigorously? 

Reviewer #1: N/A

Reviewer #2: Yes

4. Have the authors made all data underlying the findings in their manuscript fully available?

Reviewer #1: Yes

Reviewer #2: Yes

5. Is the manuscript presented in an intelligible fashion and written in standard English?

Reviewer #1: Yes

Reviewer #2: Yes

6. Review Comments to the Author

Reviewer #1: Dear authors thank you for the opportunity to review the manuscript „Extending the use of the World Health Organisations’ water sanitation and hygiene assessment tool for surveys in hospitals – from WASH-FIT to WASH-FAST”.

The need to comply with elementary hygiene rules is evident and requires no scientific investigation.

The potential application of the "tool" wash-fast seems to be an inappropriate way to counteract non-compliance with elementary hygiene rules.

Reviewer #2: Dear Authors,

thank you for your thorough revision. However, I noticed some more aspects that could again improve the manuscript:

Your conclusion section does not provide enough sufficient information. In my opinion, its length is way too short and you should claim, why WASH-FAST has an international context more detailed. Further, the main differences and improvements between WASH-FIT and WASH-FAST should be highlighted in one sentence.

Other than that, I feel your manuscript has already improved a lot.

Thank you.

7. PLOS authors have the option to publish the peer review history of their article (what does this mean?). If published, this will include your full peer review and any attached files.

Reviewer #1: No

Reviewer #2: No

---

## [Author Response · Author response to Decision Letter 1]

13 Nov 2019

Thank you for the review and comments provided for this manuscript. We have made the changes as suggested.

The Reviewers comments :

1. Dear authors thank you for the opportunity to review the manuscript „Extending the use of the World Health Organisations’ water sanitation and hygiene assessment tool for surveys in hospitals – from WASH-FIT to WASH-FAST”.

The need to comply with elementary hygiene rules is evident and requires no scientific investigation. The potential application of the "tool" wash-fast seems to be an inappropriate way to counteract non-compliance with elementary hygiene rules.

Response: 

Thank you taking time to review this manuscript and for the comment. We agree with the reviewer that it requires no scientific investigation to assert whether proper WASH is crucial– but what we are trying to do is to provide a structured tool that allows hospitals, Ministry of Health and others health agencies to evaluate compliance with hygiene rules and to summarise findings on compliance. The need for such monitoring tools is recognised by WHO and the global community who developed WASH-FIT – and the findings of their global survey of smaller facilities has been important in identifying the gap between self-evident WASH rules and what is actually in place. This evidence is being used to advocate for more support to WASH implementation.

Monitoring and reporting is an essential step to addressing and improving compliance, and an essential step for implementation of any sound public health intervention. Mechanisms for standardized monitoring and accountability of WASH in Low and Middle income countries are lacking and we therefore understand this is an important gap that needs addressing. What we have done in our work done is develop this monitoring / compliance assessment further so it is better suited for hospitals with multiple inpatient units but in addition provide a form of accountability which has been lacking in the efforts to improve WASH.

2. Reviewers Comment

Thank you for your thorough revision. However, I noticed some more aspects that could again improve the manuscript: Your conclusion section does not provide enough sufficient information. In my opinion, its length is way too short and you should claim, why WASH-FAST has an international context more detailed. Further, the main differences and improvements between WASH-FIT and WASH-FAST should be highlighted in one sentence.

Response 

Thank you for the comments. We have modified the conclusion to highlight the main differences between the WASH-FIT and WASH-FAST. In addition, we have summarized why this tool is relevant to hospitals, governments and international agencies. Page 20 Line 441-450.

Thank You 

Dr Michuki Maina 

On behalf of the Authors

---

## [Decision Letter · Decision Letter 2]

3 Dec 2019

Extending the use of the World Health Organisations' water sanitation and hygiene assessment tool for surveys in hospitals - from WASH-FIT to WASH-FAST

PONE-D-19-19422R2

Dear Dr. Maina,

We are pleased to inform you that your manuscript has been judged scientifically suitable for publication and will be formally accepted for publication once it complies with all outstanding technical requirements.

With kind regards,

Lars-Peter Kamolz, M.D., Ph.D., M.Sc.

Academic Editor

PLOS ONE

Additional Editor Comments (optional):

Reviewers' comments:

Reviewer's Responses to Questions

**Comments to the Author**

1. If the authors have adequately addressed your comments raised in a previous round of review and you feel that this manuscript is now acceptable for publication, you may indicate that here to bypass the “Comments to the Author” section, enter your conflict of interest statement in the “Confidential to Editor” section, and submit your "Accept" recommendation.

Reviewer #1: (No Response)

Reviewer #2: All comments have been addressed

2. Is the manuscript technically sound, and do the data support the conclusions?

Reviewer #1: (No Response)

Reviewer #2: Yes

3. Has the statistical analysis been performed appropriately and rigorously? 

Reviewer #1: (No Response)

Reviewer #2: Yes

4. Have the authors made all data underlying the findings in their manuscript fully available?

Reviewer #1: (No Response)

Reviewer #2: Yes

5. Is the manuscript presented in an intelligible fashion and written in standard English?

Reviewer #1: (No Response)

Reviewer #2: Yes

6. Review Comments to the Author

Reviewer #1: (No Response)

Reviewer #2: Dear authors,

thanks for your revised manuscript.

In my opinion there's a great demand for applicable monitoring tools - no further recommendations prior publication.

Thank you

7. PLOS authors have the option to publish the peer review history of their article (what does this mean?). If published, this will include your full peer review and any attached files.

Reviewer #1: No

Reviewer #2: No

---

## [Editor Report · Acceptance letter]

9 Dec 2019

PONE-D-19-19422R2 

Extending the use of the World Health Organisations’  water sanitation and hygiene assessment tool for surveys in hospitals – from WASH-FIT to WASH-FAST 

Dear Dr. Maina:

I am pleased to inform you that your manuscript has been deemed suitable for publication in PLOS ONE. Congratulations! Your manuscript is now with our production department. 

With kind regards,

on behalf of

Dr. Lars-Peter Kamolz 

Academic Editor

PLOS ONE